# Prevalence of anemia among Saudi patients with solid cancers at diagnosis in King Faisal Hospital, Taif Province, Kingdom of Saudi Arabia

**Mazen Almehmadi, Magdi Salih\*, Tariq E. Elmissbah, Abdulaziz Alsharif, Naif Alsiwiehri, Khalid Alzahrani, Alaa Shafie, Haytham Dahlawi**

Clinical Laboratory Sciences Department, College of Applied Medical Sciences, Taif University, Taif Province, Kingdom of Saudi Arabia

\* Magdi-206@hotmail.com

## Abstract

### Objectives

The aim of this study was to estimate the prevalence of anemia among patients newly diagnosed with solid malignancies at King Faisal Hospital in Taif Province, Kingdom of Saudi Arabia.

### Methods

A descriptive, cross-sectional, hospital-based study was conducted from December 2017 to March 2020. A total of 320 patients newly diagnosed with solid malignancy were examined to assess anemia prevalence.

### Results

Of 320 patients with solid cancers, 245 (76.6%) were female and 75 (23.4%) were male. The median (interquartile range) age of 57 (45 − 66) years, range between 16 and 108 years. The types of cancer included were breast (29.1%), female genital tract (20.0%), colorectal (25.3%), head and neck (10.3%), urinary bladder (4.7%), prostate (5.0%), lung (2.5%), liver (2.2%) and lymphoma (0.9%). The prevalence of anemia at diagnosis of cancer was 44.1% across all cancer types. A higher anemia prevalence was noted in colorectal (n = 46/81, 56.8%) (p = 0.047).

### Conclusion

Patients with colorectal or female genital tract cancers had a higher anemia prevalence (56.8% and 43.8%, respectively) than did patients with other cancers.

**Data Availability Statement:** All relevant data are within the paper and its Supporting Information files.

**Funding:** Dr. Mazen Almehmadi obtained funding from the deanship of scientific research project no.1-440-6143, Taif University. The funders had no role in study design, data collection and analysis, decision to publish, or preparation of the manuscript.

**Competing interests:** The authors have declared that no competing interests exist.

## Introduction

The frequency of anemia in cancer patients is high, although there is huge inconsistency in the rates reported at the time of cancer diagnosis, ranging from 39% to more than 80% [1–4], and a further 13% of nonanemic cancer patients develop anemia during the management of their malignancy [5]. Anemia is a clinical status indicated by a reduced red blood cell (RBC) mass with consequent low hemoglobin (HGB) and hematocrit amounts. Physiologically, it is defined as a reduction in the oxygen-carrying ability of the blood, which leads to tissue hypoxia [6]. Complete blood count, is used to record the number of blood cells. Anemia is reported as low levels of RBCs contained in blood and low levels of blood HGB. RBC count and HGB concentration [7]. Normal adult hematocrit values vary between 40% and 52% for men and 37% and 47% for women. Normal adult HGB values are generally 13 to 18 g/dL for men and 12 to 16.5 g/dL for women [8].

Anemia is serious in cancer patients as it may worsen health in this already frail group [9], and it has been associated with poorer medical outcomes [10, 11]. The prevalence of cancer-associated anemia varies depending on the malignancy's nature, stage, duration and spread and on the type and schedule of treatment [12, 13]. Cancer-associated anemia confers general adverse consequence mainly in anemic patients with lung, prostate, or head and neck cancers or lymphoma, which have a substantial decrease in survival time compared to their nonanemic counterparts [14]. Therefore, it is accepted that management of anemia is necessary. Sadly, suitable management is often not undertaken, which may be due to the severity of the underlying malignancy. It is essential to recognize the incidence of anemia among cancer patients to support the most appropriate treatment plans. Studies assessing the occurrence of anemia in cancer patients in the Asian context are scarce, and the precise rates in the Kingdom of Saudi Arabia are not well-established. Therefore, the aim of this study is to estimate the prevalence of anemia in solid cancer patients at diagnosis in King Faisal Hospital in the city of Taif.

## Materials and methods

### Study population

Documents were examined for patients admitted to King Faisal Hospital, Taif Province, Kingdom of Saudi Arabia, from December 2017 to March 2020 who were recently confirmed with at least one of the following cancers: breast, female genital tract, colorectal, head and neck, lung, urinary bladder, liver, prostate or lymphoma. A total of 320 patient records were identified. Inclusion criteria were as follows: male or female of any age; histopathologic confirmation of cancer; primary malignancy with no previous anticancer treatment; and presence of all data from medical history and examinations. Patients were excluded for the following reasons: a history of hematological disease or bone marrow malignancy or anemia of any cause or of chronic renal disease; or patient had received a blood transfusion or was on follow-up for chemotherapy, radiotherapy or surgery.

### Data collection

Data on age, gender, tumor diagnosis and blood cell tests were examined. Anemia was defined as HGB concentration <11.5 g/dL for females and <13.0 g/dL for males [15]. Based on this measurement, study groups were divided into two main groups: anemic and nonanemic. All study participants were measured for RBC, HGB and PCV. Parameters were determined using the hematology analyzer Cell-Dyn 1800 (Abbott Laboratories Diagnostics Division, USA). Performance of the hematology analyzer was controlled by running quality control measures alongside the study participants' samples.

## Statistical analysis

Data were entered in computer using SPSS 16.0 software (SPSS Inc., Chicago, IL) for Windows continuous data (age, HB, RBC and PCV) were checked for normality using Shapiro Wilk test. Variables were not normally distributed and were shown as median (interquartile) and Kruskal–Wallis test was used to compare between different tumor types. Proportions were compared by Chi square test. P value < 0.05 was considered to be significant.

## Ethical considerations

The study received ethical approval from the Directorate of Health Affairs in Taif, registration number HAP-02-T-067, Taif, Kingdom of Saudi Arabia. All patient personal data were fully anonymized.

## Results

A total of 320 cancer patients were analyzed: 245 (76.6%) females and 75 (23.4%) males. In terms of age, most of the study group (129, 40.3%) fell in the 56–74 years group, with the age range 16 to 108 years, a median (interquartile range) age of 57 (45 — 66) years. The types of cancer included were breast (29.1%), female genital tract (20.0%), colorectal (25.3%), head and neck (10.3%), urinary bladder (4.7%), prostate (5.0%), lung (2.5%), liver (2.2%) and lymphoma (0.9%). The HGB levels ranged from 4.8 g/dL to 18.8 g/dL with a median (interquartile range) of 12.9 (11.0 — 14.3) g/dL (Fig 1).

The median (interquartile range) HGB for male patients was 12.100 (6.6–17.3) g/dL and for female patients was 13.00 (4.8–18.8) g/dL. Anemia was identified in 141 (44.1%), of the 320 patients and median (interquartile) concentration of HGB was 10.5 g/dL in these anemic patients, whereas it was 14.2 g/dL in the 179 nonanemic patients. Generally, the prevalence of anemia at cancer diagnosis was 44.1% across cancer types, and higher anemia prevalence was noted in colorectal (n = 46/81, 56.8%) (OR = 0.60; 95% CI 0.36–0.98) ($p$ = 0.047), (Fig 2 and Table 1).

Female patients had a higher median (interquartile range) HGB level than male patients (13.00 (11.4 —14.3) g/dL for female and 12.10 (10.3 —14.4) g/dL for male, $p$ = 0.265), a higher median (interquartile range) RBC count (4.81(4.25 — 5.25) •$10^{12}$/L for female and 4.70 (4.12 — 5.19)•$10^{12}$/L for male, $p$ = 0.293) and a higher median (interquartile range) PCV value (39.60 (35.15 — 43.80)% for female and for 36.90 (31.90 — 44.00)% male, $p$ = 0.125). In terms of cancer types, patients with breast cancer showed the highest median (interquartile range) HGB level (13.50 (12.40 — 14.75) g/dL). The median (interquartile range) RBC count was also highest in breast cancer patients (4.92 (4.49 — 5.26)•$10^{12}$/L), and no patient with breast cancer had a PCV level lower than 13.90% (range 13.90–51.90%). In terms of lowest median (interquartile range) values, urinary bladder cancer patients had the lowest median (interquartile range) RBC (4.39 (3.97 — 4.77)•$10^{12}$/L), lymphoma patients had the lowest median (interquartile range) HGB concentration (10.30 (9.50 — 0.00)g/dL) and PCV value (33.50 (28.10 — 0.00)%) (Table 2).

## Discussion

Anemia in cancer patients is a consequence of the malignant tumor, anticancer therapy, bleeding, malnutrition, hemolysis, or endocrine syndromes. In the present study, 320 treatment-naive, recently diagnosed solid cancer patients at King Faisal Hospital, Taif Province, Kingdom of Saudi Arabia, were assessed for anemia. This study found the frequency of anemia in these patients across tumor types was 44.1%, which is much higher than the rates in China (18.98%),

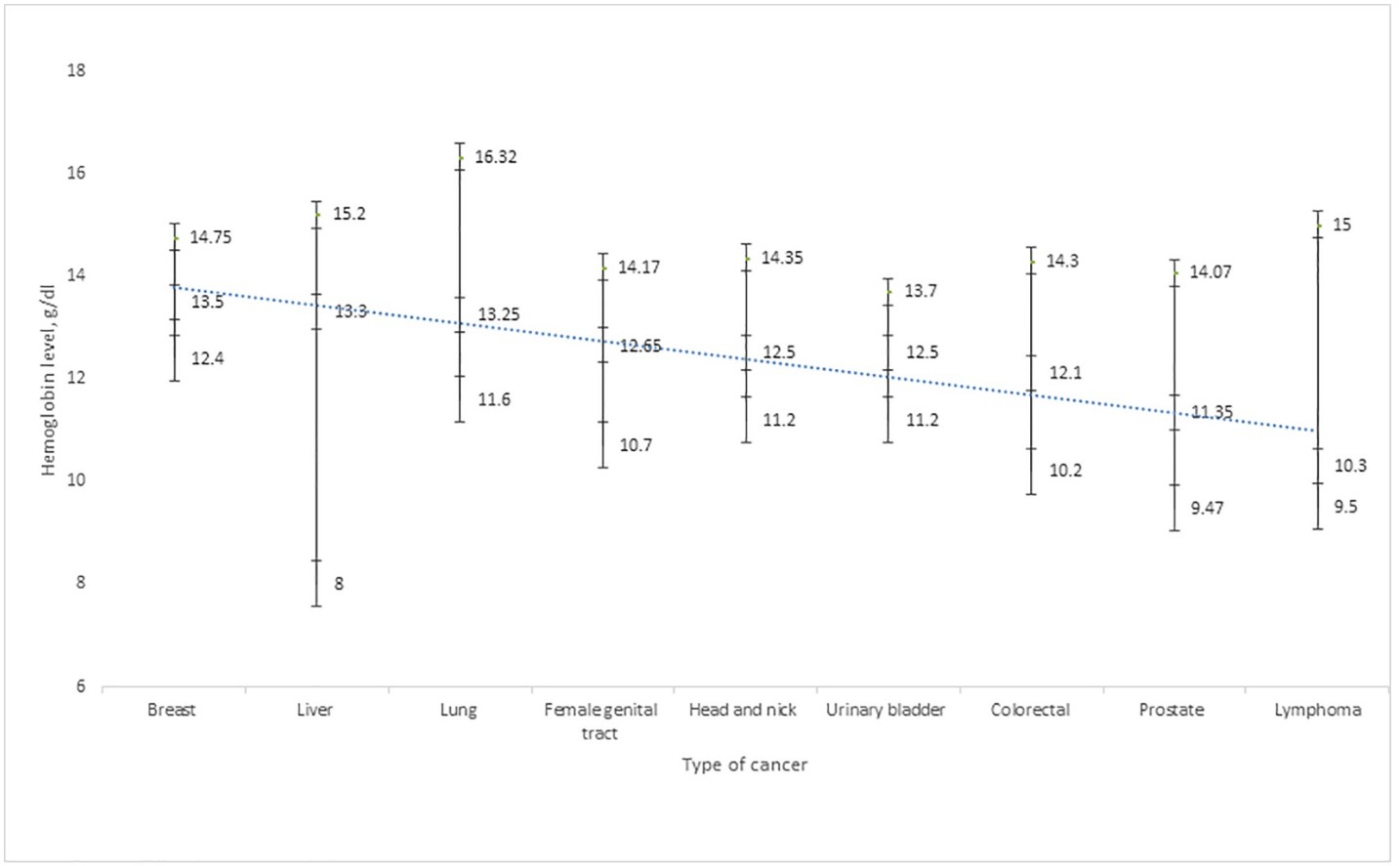

**Fig 1. The hemoglobin levels and cancer types.**

but only marginally higher than Australia (39.3%), the US (35%), and Thailand (41%) [16–19]. Nevertheless, our result is lower than the rates found by researchers in Europe (54.4%), India (54.7%) and Belgium (55.7%) [20–22]. The variation of frequencies in our study is a result of differences in the description of anemia, the study group composition and the investigation period.

The most common cancers noted in our study were breast (93/320, 29.1%) followed by colorectal cancer (81/320, 25.3%). Our results are similar to those from studies conducted in Thailand and Ethiopia, where breast cancer (26.2%) and colorectal cancer (26.7%) were the leading cancers among the detected tumor varieties [21], respectively. The prevalence of anemia varied by tumor type, with our study demonstrating that 68.8%, 56.8% and 43.8% of prostate, colorectal and female genital tract cancer patients, respectively, were anemic. These rates are lower than those in the US, 78% of prostate cancer patients were reported as anemic; in Norway, 74.7% of colorectal cancer patients were reported as anemic; and in Australia, 65% of female genital tract cancers patients were anemic at admission [23–26]. The differences in rates between our study and these other studies may result from differences in the characterization of anemia and differences in study strategies used. Female patients and patients older than 56 years constitute more than 50% of anemic study group. We found comparable results in China, Belgium and Sudan [16, 22, 27]. In this study, females were more likely to be anemic than males because female genital tract and breast cancers accounted for nearly half of the reported cancer cases. One of the complications of female genital tract cancer is bleeding,

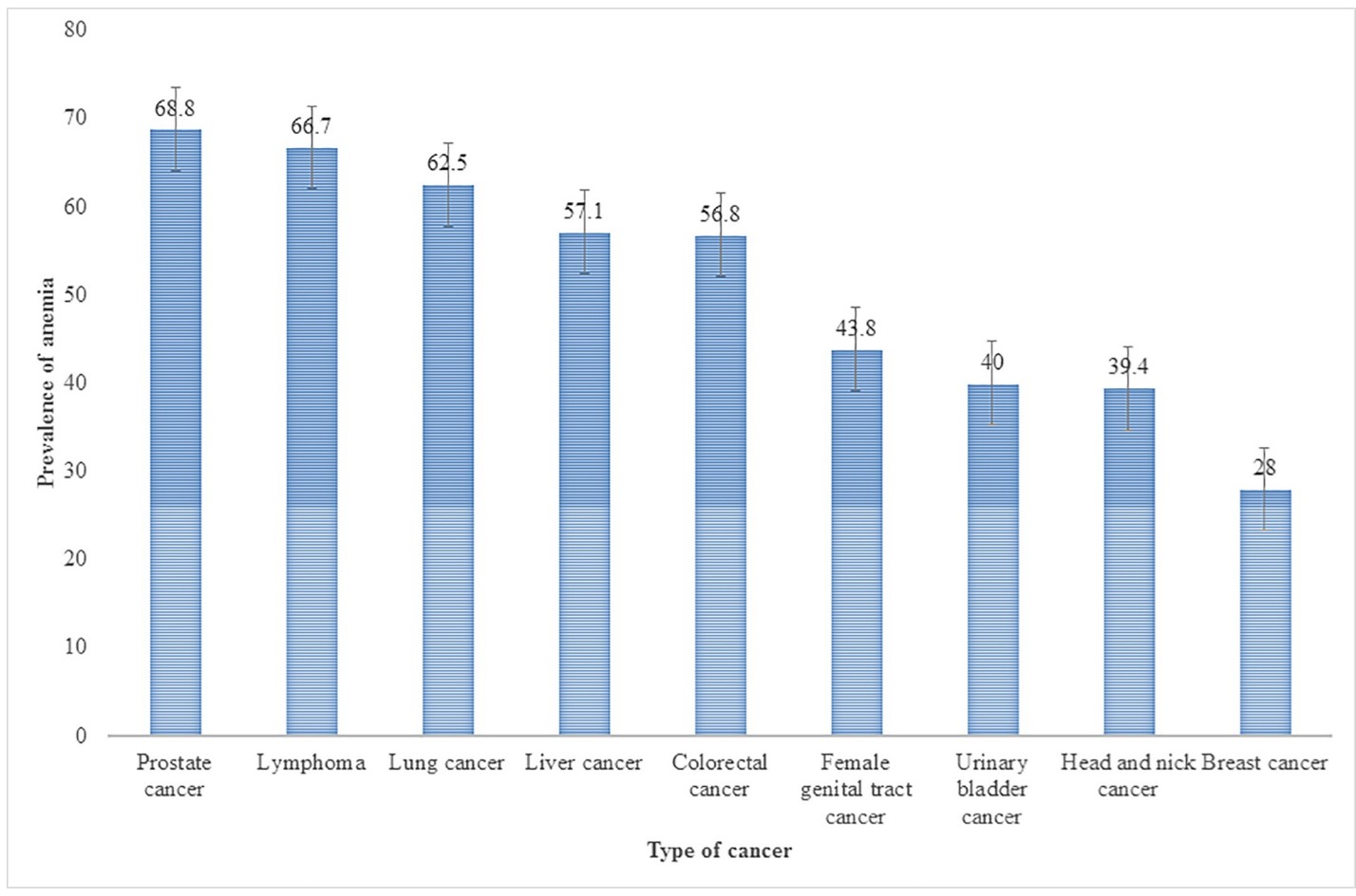

**Fig 2. Prevalence of anemia among cancer types.**

**Table 1. Prevalence of anemia and factors among newly diagnosed solid cancer patients at King Feisal Hospital at Taif Provinces Kingdom of Saudi Arabia during December 2017 to March 2020 (n = 320).**

|  | N (%) | N (%) | Odds ratio (95% Confidence) | Chi-Square |
|---|---|---|---|---|
| Tumor types | Anaemic | Non anaemic |  | P value |
|  | (N = 141) | (n = 179) |  |  |
| Breast cancer | 26 (28.0) | 67 (72.0) | 2.02 (1.23 — 3.39) | <0.001 |
| Female genital tract cancer | 28 (43.8) | 36 (56.2) | 1.01 (0.58 — 1.75) | 0.955 |
| Colorectal cancer | 46 (56.8) | 35 (43.2) | 0.60 (0.36 — 0.98) | 0.047 |
| Head and nick cancer | 13 (39.4) | 20 (60.6) | 1.21 (0.58 — 2.58) | 0.568 |
| Lung cancer | 5 (62.5) | 3 (37.5) | 0.47 (0.09 — 2.08) | 0.288 |
| Urinary bladder cancer | 6 (40.0) | 9 (60.0) | 1.18 (0.40 — 3.64) | 0.745 |
| Liver cancer | 4 (57.1) | 3 (42.9) | 0.59 (0.10 — 2.90) | 0.896 |
| Prostate cancer | 11 (68.8) | 5 (31.2) | 0.36 (0.11 — 1.04) | 0.052 |
| Lymphoma | 2 (66.7) | 1 (33.3) | 0.39 (0.01 — 5.23) | 0.432 |
| **Gender** |  |  |  |  |
| Female | 94 (38.4) | 151 (61.6) | 1.26 (0.90 — 1.78) | 0.173 |
| Male | 47 (62.7) | 28 (37.3) | 0.47 (0.28 — 0.79) | 0.003 |

**Table 2. Cancer types, categorization based on RBC count.** Hb amount and PCV% at King Feisal Hospital at Taif Provinces (N = 320).

| Cancer types | | RBC m/μL | Hb g/dl | PCV% |
|---|---|---|---|---|
| | N (%) | Median (25% ─ 75%) | Median (25% ─ 75%) | Median (25% ─ 75%) |
| Breast | 93 (29.1%) | 4.92 (4.49 ─ 5.26) | 13.50 (12.40 ─ 14.75) | 41.50 (38.40 ─ 44.65) |
| Female genital tract | 64 (20.0%) | 4.83 (4.15 ─ 5.35) | 12.65 (10.70 ─ 14.17) | 39.40 (33.32 ─ 43.72) |
| Colorectal | 81 (25.3%) | 4.64 (4.15 ─ 5.14) | 12.10 (10.20 ─ 14.30) | 35.90(30.70 ─ 42.85) |
| Head and nick | 33 (10.3%) | 4.72 (4.21 ─ 5.18) | 12.50 (11.20 ─ 14.35) | 40.40 (34.65 ─ 44.70) |
| Lung | 8 (2.5%) | 4.59 (4.29 ─ 6.02) | 13.25 (11.60 ─ 16.32) | 40.75 (34.75 ─ 48.95) |
| Urinary bladder | 15 (4.7%) | 4.39 (3.97 ─ 4.77) | 12.50 (11.20 ─ 13.70) | 37.10 (32.80 ─ 40.30) |
| Liver | 7 (2.2%) | 5.14 (3.31 ─ 5.21) | 13.30 (8.00 ─ 15.20) | 38.20 (25.10 ─ 47.10) |
| Prostate | 16 (5.0%) | 4.97 (3.60 ─ 5.59) | 11.35 (9.47 ─ 14.07) | 38.10 (29.72 ─ 43.20) |
| Lymphoma | 3 (0.9%) | 3.74 (3.26 ─ 0.00) | 10.30 (9.50 ─ 0.00) | 33.50 (28.10 ─ 0.00) |
| Total | 320 (100%) | Kruskal Wallis test P value 0.057 | Kruskal Wallis test P value = 0.018 | Kruskal Wallis test P value = 0.003 |

Red blood cell count (RBC). Hemoglobin level (Hb). Packed cell volume (PCV). Number (N). Percentage (%). Interquartile Range (25% ─ 75%).

which can lead to anemia or at least to lowered levels of HGB subsequent to iron deficiency. In the present study, this might be the most common cause of anemia, as anemia caused by breast cancer only is doubtful. The main likely cause for the higher anemia percentage among older patients compared to younger patients is the set of functional alterations that occur as one ages. These lead, for instance, to a degeneration in hematopoietic stem cell reserves and proliferation ability, which leads to suppression of erythropoiesis. Anemia incidence also varied by tumor type. Higher anemia rates were seen in colorectal and female genital tract carcinomas (32.6% and 19.8%, respectively). The conceivable primary explanation for this result in colorectal tumors is that the disturbance of digestive function led to unseen and long-duration blood loss [16]. The likely reason for the high rate among gynecologic cancer patients is that the main complication of female genital tract cancer is vaginal bleeding.

We found that a significant number of breast and colorectal cancer patients were anemic in terms of HGB concentration, RBC count, and PCV, but for most of the other cancer patients, these values fell within the normal range of reference values. These outcomes may reflect the normal erythropoietic activity in bone marrow and probably exclude the decreased erythropoietin response and common malnutrition as the leading causes of anemia in solid cancer patients at diagnosis. Therefore, reductions in HGB level, RBC count, and PCV in solid cancer patients may be a result of tumor-induced hemolysis. Bhattathiri showed the relation of red cell indices in oral cancer with clinical staging but did not examine the other solid cancer types. Bhattathiri studied 217 patients with oral cancer and reported decreases in mean HGB level (by 63%), RBC count (43%) and PCV (48.4%) [27].

To our knowledge, this study is the first to assess erythrocyte indices in association with anemia among solid cancer patients at diagnosis in the kingdom.

## Limitations

The generalizability of our results may be restricted by the small and inadequate numbers in each cancer-type cluster, which resulted from the inclusion of only newly diagnosed cancers. Shortage of accompanying iron studies, blood cell morphology assessment to characterize anemia types are other limitations of this study we recommend further studies into the impact of anemia on cancer grade and management outcome using cohort study.

## Conclusion

In this study, the general prevalence of anemia across tumor types was 44.1%. In considering specific cancer types, patients with colorectal and female genital tract cancers recorded higher anemia prevalence (32.6% and 19.8%, respectively) than patients with other cancers. Female patients and patients $\geq$55 years old showed higher frequencies of anemia than male patients and patients <55 years.

## Supporting information

**S1 File. Anemia solid cancers data.**
(SAV)

## Acknowledgments

We would like to express our deep gratitude to Professor Ishag Adam for his advice and assistance in data analysis, and special appreciation to all medical staff in King Faisal hospital for assistance in data collection and lab works.

## Author Contributions

**Conceptualization:** Magdi Salih, Tariq E. Elmissbah, Alaa Shafie, Haytham Dahlawi.

**Data curation:** Mazen Almehmadi, Magdi Salih.

**Formal analysis:** Magdi Salih.

**Investigation:** Abdulaziz Alsharif.

**Methodology:** Tariq E. Elmissbah, Khalid Alzahrani.

**Project administration:** Khalid Alzahrani.

**Resources:** Mazen Almehmadi, Abdulaziz Alsharif, Haytham Dahlawi.

**Software:** Naif Alsiwiehri.

**Validation:** Naif Alsiwiehri, Alaa Shafie.

**Visualization:** Tariq E. Elmissbah.

**Writing – original draft:** Magdi Salih.

**Writing – review & editing:** Magdi Salih.

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
