## [Decision Letter · Decision Letter 0]

18 Nov 2020

PONE-D-20-28573

Prevalence of Anemia among Saudi Patients with Solid Cancers at Diagnosis in King Faisal Hospital, Taif Province, Kingdom of Saudi Arabia

PLOS ONE

Dear Dr. Salih,

Thank you for submitting your manuscript to PLOS ONE. After careful consideration, we feel that it has merit but does not fully meet PLOS ONE’s publication criteria as it currently stands. Therefore, we invite you to submit a revised version of the manuscript that addresses the points raised during the review process.

Specifically, concerns were raised over the statistical analysis, and both reviewers had suggestions to improve the readability of the manuscript.

We look forward to receiving your revised manuscript.

Kind regards,

Colin Johnson, Ph.D.

Academic Editor

PLOS ONE

Journal Requirements:

2. Please ensure that you include a title page within your main document. You should list all authors and all affiliations as per our author instructions and clearly indicate the corresponding author.

3. Please include your tables as part of your main manuscript and remove the individual files. Please note that supplementary tables (should remain/ be uploaded) as separate "supporting information" files

Reviewers' comments:

Reviewer's Responses to Questions

**Comments to the Author**

1. Is the manuscript technically sound, and do the data support the conclusions?

Reviewer #1: Yes

Reviewer #2: Yes

2. Has the statistical analysis been performed appropriately and rigorously? 

Reviewer #1: No

Reviewer #2: Yes

3. Have the authors made all data underlying the findings in their manuscript fully available?

Reviewer #1: Yes

Reviewer #2: No

4. Is the manuscript presented in an intelligible fashion and written in standard English?

Reviewer #1: Yes

Reviewer #2: Yes

5. Review Comments to the Author

Reviewer #1: Comments to the author,

Abstract section,

Line 7- change is to "was". In your objective... newly diagnosed... Does this indicate your study to be prospective?

Line 12- the term incidence preferably good to be replaced by prevalence

Line 15- Add range of your study participants to the age groups

Line 20- P-value in the absence of OR and/or Chi square does not have helpful meaning. So, here the analysis and result should be re-analyzed with the help of OR, chi square and 95% C.I to see association between the variables and to determine clinical significance.

Introduction section,

Line 38-40: the term" erythrocytosis" seems not necessary in this paper as it is not relevant to the topic of study

Line 46: replace adversarial with the word" adverse"

Under Materials and methods section,

What is your actual study population? Is your study participants patient or medical records? How do you process your ethical assurance?

Line 69: You did not cite for referencing cut off value for anemia and again it lacks in reference section

How did you get 320 of your sample size? Sample size calculation is not stated. What type of sampling technique did you use to collect your sample?

What is your study design? Is it retrospective or prospective?

In your analysis part,

Line 84: Even if you mentioned some analytical parameters like Chi square and ANOVA test, it lacks applicability in your findings. What is the value of using ANOVA test?

In result section,

Line 94: 56-74 years

Line 106: What is the operational definition base line? Is it consistent with your objective?

Generally, it is not well stated. Only descriptive statistics were used. Majorly, OR and or Chi-square were missed; re-analysis should be operated especially in Table 1 &2. Even the existing p-value is very vague. I t is not clear how p-value is analyzed. For each variable, there should be p-value except constant variable. Again in Table 1 with gender variable, p-value=0.00. By any means, p-value should no be zero. So, re-construct these tables using OR and/or Chi-square, 95%C.I and/or p-value for each independent variable except the constant one.

Discussion part,

Line 125 & 131: The same reference is cited [20]; in fact it is not the same. Check your citation and references, too for correction.

Line 125-127; 136-138 & 151-153: Unless there is reference for your possible reason of the differences between your findings and others, it seems personal judgment, which is not scientifically acceptable.

Line 164: It is not clear. Does the statement belong to your finding?

In reference section,

Ref No 20 & 22 are the same. Could you address the issue?

Reviewer #2: This study analyzed the prevalence of anemia in newly diagnosed cancer patients in Saudi Arabia. The number of subjects is sufficient to drive the intended analysis. I have the following comments:

1. The introduction should be shortened, eliminating the extended defintiion of anemia which I assume is very familiar to the PLOS ONE audience. I suggest keeping it simple and short, adopting WHO definition (Hb levels).

2. In methods, the division of RBC parameters in "normal", "low" and "high" groups seems pointless and dispensable as it is not used nor discussed throughout the manuscript.

3. In Results, some data presented in the second paragraph are the same reported in the first paragraph, and should be removed.

4. The prevalence of anemia in prostate cancer and lymphoma patients should be cautiously described as the number of subjects was very low in both groups, making it unnapropriate to assume a true high prevalence of anemia in these patients.

5. Please state the unit for frequency in figures 1 and 2 (number or percentage … I assume those are numbers, correct?).

6. I suggest a reformulation of table 2 with respect to the cut-off points adopted for RBC, Hb and PCV, as they were not the same adopted for anemia definition under “Methods”, and should also be different between male and female patients.

6. PLOS authors have the option to publish the peer review history of their article (what does this mean?). If published, this will include your full peer review and any attached files.

Reviewer #1: No

Reviewer #2: No

---

## [Author Response · Author response to Decision Letter 0]

30 Dec 2020

Response to the Comments of Reviewers

December 15, 2020

Subject: Revision and resubmission of manuscript PONE-D-20-28573

Prevalence of Anemia among Saudi Patients with Solid Cancers at Diagnosis in King Faisal Hospital, Taif Province, Kingdom of Saudi Arabia

Dear Dr. Colin Johnson,

Thank you for your letter and the opportunity to revise our paper. The suggestions offered by the reviewers have been immensely helpful. I have included the reviewer comments immediately after this letter and responded to them individually, indicating exactly how we addressed each concern or problem and describing the changes we have made. The revisions have been approved by all authors. The changes are marked in yellow in the paper, and the revised manuscript will be uploaded.

Reviewer #1: Comments to the author,

Abstract section,

Comment. Line 7- change is to "was". In your objective... newly diagnosed... Does this indicate your study to be prospective?

Respond: Thank you for your assessment within line7 the suggested correction has been made.

No we mean that our study group were newly diagnosed cancer patient and not subjected to cancer therapy as it has complications that my affects our results 

Comment. Line 12- the term incidence preferably good to be replaced by prevalence

 Respond: thank you for this excellent observation for line12 the suggested correction has been made and we replaced the word by prevalence.

Comment. Line 15- Add range of your study participants to the age groups

Respond: Thank you! We found your comments helpful and have revised accordingly the suggested correction has been made, and range of study participants to the age groups added 

Comment. Line 20- P-value in the absence of OR and/or Chi square does not have helpful meaning. So, here the analysis and result should be re-analyzed with the help of OR, chi square and 95% C.I to see association between the variables and to determine clinical significance.

Introduction section,

Respond 

We agree with the reviewer’s assessment of the analysis. Our chi square does make it difficult to fully interpret the data. In addition, in its current form, we agree it would be hard to tell that this measurement constitutes a significant improvement over previously reported values. we reanalyzed our data starting with normality checking using the Shapiro-Wilk’s W test to determine whether our data is normally distributed, the checks showed Value < 0.05 suggest that our data is not normally distributed so that we used the non-parametric tests the Kruskal Walis that is non-parametric test to determines whether the medians and interquartile range of the Hb, PCV and Red blood cell count are different to determine the presence of anemia based on these parameters, and we compare it to different cancers types. We have therefore re-analyzed the data using odds ratio, chi square and 95% C.I as suggested with reviewer.” Please see re-constructed table 1& 2.

Comment. Line 38-40: the term" erythrocytosis" seems not necessary in this paper as it is not relevant to the topic of study

Respond 

Thank you very much erythrocytosis" definition was deleted 

Comment. Line 46: replace adversarial with the word" adverse" Under Materials and methods section,

Respond 

Thank you for comment the word adversarial was replaced with the “adverse"

Comment. What is your actual study population? Is your study participants patient or medical records? How do you process your ethical assurance?

Respond

Thank you very much the actual study populations “were medical records and we processed our ethical assurance by taking permission from local health authorities and pathology department at the hospital (The study received ethical approval from the Directorate of Health Affairs in Taif, registration number HAP-02-T-067, Taif, Kingdom of Saudi Arabia. All patient personal data were fully anonymized)

Comment. Line 69: You did not cite for referencing cut off value for anemia and again it lacks in reference section

Respond 

Thank you very much with regard to cite for referencing cut off value for anemia we add the missing reference its number 15 in reference section (its Beutler E, Waalen J. The definition of anemia: what is the lower limit of normal of the blood hemoglobin concentration?. Blood. 2006;107(5):1747-1750. doi:10.1182/blood-2005-07-3046)

Comment. How did you get 320 of your sample size? Sample size calculation is not stated. What type of sampling technique did you use to collect your sample?

Respond 

A sample size of 320 patients was calculated based on the equation and we assumed that 29.5% of the patients with cancer would have anemia. 

N= z2*p*(1-p)/e2

N= sample size for infinite population

z= Z score. We considered confidence level 95% then Z score=1.96.

p= % of population probability (assumed to be 29.5%=0. 29.5)

e= desired margin of error. It was taken 5%=0.05.

Comment. What is your study design? Is it retrospective or prospective?

In your analysis part,

Respond 

Thank you its retrospective study 

Comment. Line 84: Even if you mentioned some analytical parameters like Chi square and ANOVA test, it lacks applicability in your findings. What is the value of using ANOVA test?

In result section,

Respond 

Thank you very much and we are sorry for this confusion we agree with the reviewer and reanalyzed the data and deleted this confusion, instead of the ANOVA The Kruskal Wallis test was used the test determines whether the medians of two or more groups are different we explain this above .

Comment.Line 94: 56-74 years

Respond 

Thank you the suggested correction has been made

Comment. Line 106: What is the operational definition base line? Is it consistent with your objective?

Generally, it is not well stated. Only descriptive statistics were used. Majorly, OR and or Chi-square were missed; re-analysis should be operated especially in Table 1 &2. Even the existing p-value is very vague. I t is not clear how p-value is analyzed. For each variable, there should be p-value except constant variable. Again in Table 1 with gender variable, p-value=0.00. By any means, p-value should no be zero. So, re-construct these tables using OR and/or Chi-square, 95%C.I and/or p-value for each independent variable except the constant one.

Respond 

Thank you so much for comment. In general, a baseline is a well-defined, well-documented reference that serves as the foundation for different assessment, the operational definition of anemia depends on Hemoglobin levels at baseline an individual with baseline Hb concentration at the higher end of normal, decrease in Hb concentration to the lower end of normal might be considered anemic and we used this baseline for classification of anemia in our study group and this liked to the objective of this study.

Regarding statistics comment we agree with reviewer so that we re-analyzed our data we include the OR, chi square and 95% C.I and the association between the anaemia and different types of cancers was determine and we have re-analyzed the data presented in former tables and re-constructed table 1& 2 that include p-value For each variable. 

Discussion part,

Line 125 & 131: The same reference is cited [20]; in fact it is not the same. Check your citation and references, too for correction.

Respond 

Thank you so much the references citation was checked and corrected 

Line 125-127; 136-138 & 151-153: Unless there is reference for your possible reason of the differences between your findings and others, it seems personal judgment, which is not scientifically acceptable.

Respond 

Thank you so much it’s not personal judgment for our finding in lines 125-127, is supported with well-known scientific information that description of anemia, the study group composition may affects the outcomes 136-138 we cited comparable results from China, Belgium and Sudan please see references numbers (16,22 and 26) and for 151-153 data its eminent that one of the most common causative factor of anemia among female is bleeding either from menstrual cycle or female genital tract tumors 

Line 164: It is not clear. Does the statement belong to your finding?

Respond: Thank you so much yes the statement describe our study finding 

In reference section,

Ref No 20 & 22 are the same. Could you address the issue?

Respond: Thank you so much for catching these glaring and confusing errors, which we have now corrected in the text and deleted the duplicated reflectance number 22 and rearranged the whole references 

Reviewer #2: This study analyzed the prevalence of anemia in newly diagnosed cancer patients in Saudi Arabia. The number of subjects is sufficient to drive the intended analysis. I have the following comments:

1. The introduction should be shortened, eliminating the extended definition of anemia which I assume is very familiar to the PLOS ONE audience. I suggest keeping it simple and short, adopting WHO definition (Hb levels).

Respond

Thank you for reminding us how important it is to present short and scripts in a concise introduction. We agree that definition of anemia is very familiar to readers of journal however onetime we need to start from basic point to recognize our hypothesis.

Changes: We shorten and simplify anemia definition and agreeing WHO meaning. We believe this sets the information out clearly and comparatively and is a format that readers will return to when seeking information on the manuscript’s uncomplicated idea. The changes in text appear in the revised paper.

2. In methods, the division of RBC parameters in "normal", "low" and "high" groups seems pointless and superfluous as it is not used nor discussed throughout the manuscript.

Respond: Thank you so much the reviewer makes a great point: and this division may appear superfluous and confusing to reader so we remove this part from the method section in the revised manuscript.

3. In Results, some data presented in the second paragraph are the same reported in the first paragraph, and should be removed.

Respond: Thank you so much as suggested, the data presented in results have been revised in order to more effectively convey the central idea of the manuscript. 

4. The prevalence of anemia in prostate cancer and lymphoma patients should be cautiously described as the number of subjects was very low in both groups, making it inappropriate to assume a true high prevalence of anemia in these patients.

Respond. We agree with the reviewer assessment of lymphoma patients and prostate cancer findings unfitting to adopt high prevalence of anemia in these patients for low number and it’s impossible to fully interpret the data in terms of the prevailing theories the reformed manuscript was revised accordingly. 

5. Please state the unit for frequency in figures 1 and 2 (number or percentage … I assume those are numbers, correct?).

Respond. Thank you very much as suggested by the reviewer, we have revised Figure 1 and 2 we reconstructed these figures 

6. I suggest a reformulation of table 2 with respect to the cut-off points adopted for RBC, Hb and PCV, as they were not the same adopted for anemia definition under “Methods”, and should also be different between male and female patients.

Respond:

We have re-analyzed the data using odds ratio, chi square and 95% C.I and re-constructed table 1& 2 I hope it’s clearer now

We hope the revised manuscript will better suit PLOS Journal but are happy to consider further revisions, and we thank you for your continued interest in our research.

Sincerely,

Dr. Magdi Mansour Salih 

Correspondent author

---

## [Decision Letter · Decision Letter 1]

15 Jan 2021

Prevalence of Anemia among Saudi Patients with Solid Cancers at Diagnosis in King Faisal Hospital, Taif Province, Kingdom of Saudi Arabia

PONE-D-20-28573R1

Dear Dr. Salih,

We’re pleased to inform you that your manuscript has been judged scientifically suitable for publication and will be formally accepted for publication once it meets all outstanding technical requirements.

Kind regards,

Colin Johnson, Ph.D.

Academic Editor

PLOS ONE

Additional Editor Comments (optional):

Reviewers' comments:

Reviewer's Responses to Questions

**Comments to the Author**

1. If the authors have adequately addressed your comments raised in a previous round of review and you feel that this manuscript is now acceptable for publication, you may indicate that here to bypass the “Comments to the Author” section, enter your conflict of interest statement in the “Confidential to Editor” section, and submit your "Accept" recommendation.

Reviewer #1: All comments have been addressed

Reviewer #2: All comments have been addressed

2. Is the manuscript technically sound, and do the data support the conclusions?

Reviewer #1: Yes

Reviewer #2: (No Response)

3. Has the statistical analysis been performed appropriately and rigorously? 

Reviewer #1: Yes

Reviewer #2: (No Response)

4. Have the authors made all data underlying the findings in their manuscript fully available?

Reviewer #1: Yes

Reviewer #2: (No Response)

5. Is the manuscript presented in an intelligible fashion and written in standard English?

Reviewer #1: Yes

Reviewer #2: (No Response)

6. Review Comments to the Author

Reviewer #1: I want to appreciate the author in charge of this manuscript for addressing almost all of my comments. So, I recommend this paper to be published even if the decision is up to the editor-in chief. Thank you again.

Reviewer #2: (No Response)

7. PLOS authors have the option to publish the peer review history of their article (what does this mean?). If published, this will include your full peer review and any attached files.

Reviewer #1: No

Reviewer #2: No

---

## [Editor Report · Acceptance letter]

19 Jan 2021

PONE-D-20-28573R1 

Prevalence of Anemia among Saudi Patients with Solid Cancers at Diagnosis in King Faisal Hospital, Taif Province, Kingdom of Saudi Arabia 

Dear Dr. Salih:

I'm pleased to inform you that your manuscript has been deemed suitable for publication in PLOS ONE. Congratulations! Your manuscript is now with our production department. 

Kind regards, 

on behalf of

Dr. Colin Johnson 

Academic Editor

PLOS ONE